# Industrialization drives the gut microbiome and resistome of the Chinese populations

Chen Tian,[1] Tongzuo Zhang,[2] Daohua Zhuang,[1] Yu Luo,[1] Teng Li,[1] Fangfang Zhao,[3] Jianan Sang,[1] Zecheng Tang,[1] Peicheng Jiang,[1] Tao Zhang,[1] Pengfei Liu,[4,5,6] Lei Zhu,[1] Zhigang Zhang[1]

**ABSTRACT** Industrialization has driven lifestyle changes in eastern and western Chinese populations, yet we have a poor understanding of the dynamic changes in their gut microbiome and resistome under industrialization, which is essential for the scientific management of public health. Here, this study employed metagenomics to analyze the gut microbiota of 1,382 healthy individuals from China, including 415 individuals from the eastern region of advanced industrialization and 967 individuals from the western region of developing industrialization. Compared with western populations, eastern populations show a significant increase in interindividual dissimilarity of microbial species composition and metabolic pathways but a significant decrease in intraindividual species and functional diversity. Furthermore, our results found significantly less abundance and richness of antibiotic resistance genes (ARGs) in the gut microbiota of eastern populations, alongside a lower prevalence of unique core ARG subtypes. For the 12 core ARG types shared between eastern and western populations, the mean relative abundance of two types was notably higher in the eastern populations, while eight core ARG types had significantly higher mean relative abundance in the western populations. Based on the reconstruction of metagenomic assembled genomes, we found that *Escherichia coli* genomes from western populations carried more virulence factor genes (VFGs) and mobile genetic elements (MGEs) compared to those from eastern populations. This large-scale study for the first time revealed industrialization potentially led to unexpected alterations of the gut microbiome and resistome between eastern and western populations that provide a vital implication for Chinese public health and may aid in the development of region-specific strategies for managing pathogenic infections.

**IMPORTANCE** As China experiences rapid but uneven industrialization, understanding its effect on people's gut bacteria is critical for public health. This study reveals how industrialization may reshape the health risks related to gut bacteria and antibiotic resistance. This work provides crucial information to help create customized public health policies for different regions.

**KEYWORDS** industrialization, public health, gut microbiome, antibiotic resistance gene

The human gut harbors highly diverse microbial communities that exhibit variations across the human societies studied to date, with the most significant changes being associated with levels of industrialization (1). Westernized lifestyles are linked to decreased bacterial diversity and functional alterations (2, 3). These alterations could serve as causative or contributing factors to the increasing prevalence of non-communicable chronic diseases (NCCDs) in industrialized societies (4). With the rise in antibiotic usage, the microbiota of industrialized societies may adapt more effectively to the industrialized host lifestyle by developing increased resistance to antibiotics (5). Antimicrobial resistance (AMR) has caused 4.95 million deaths in 2019 (6). If no action

**Peer Reviewer** Yongqin Liu, Lanzhou University, Lanzhou, China

Address correspondence to Zhigang Zhang, zhangzhigang@ynu.edu.cn, Pengfei Liu, liupf@lzu.edu.cn, or Lei Zhu, zhulei_evan@126.com.

Chen Tian, Tongzuo Zhang, Daohua Zhuang, and Yu Luo contributed equally to this article. Author order was determined by their contribution to the article.

The authors declare no conflict of interest.

**Ed. Note:** A conflict of interest was identified after acceptance of this paper, and the senior editor provided an additional final review of the paper.

is taken, 10 million people are expected to die from antibiotic resistance by 2050, with potential economic losses reaching $100 trillion (7). AMR is a major challenge to achieving One Health as antibiotic resistance genes (ARGs) can spread and evolve among humans, animals, and the environment (8).

China is facing a rapid industrialization process, but development is uneven between the eastern and western regions. Industrialization is accelerating in eastern China, while the degree of industrialization in the western region is relatively low. China is also experiencing a series of problems faced by global industrialization. People in eastern China have a westernized diet and consume more antibiotics (9, 10), while people in most areas of western China follow a traditional lifestyle. It is like the phenomenon of uncoordinated regional development caused by global industrialization. According to the survey, the average discharge density of antibiotics in river basins in eastern and southern China is more than six times that in river basins in western China (10). Through the monitoring of drug resistance genes in urban sewage in China, it was found that the ARG burden is significantly higher in eastern China, showing 1–2 orders of magnitude higher than in western China (11). The map of global soil ARGs shows that eastern China is a hotspot for ARGs (12). The average richness and absolute abundance of ARGs in lagoon sediments in Xiamen, China, are 11 times and 53 times higher than those in lake sediments in Tibet, respectively (13). The absolute abundance of ARGs in cow manure from cattle farms in Shandong is about 100 times higher than that in Shaanxi (14). Environmental and animal studies show substantial differences in antibiotic pressure between eastern and western China. However, as industrialization continues to advance, the alterations in the gut microbiome and resistome among the populations between eastern and western China remain largely unexplored. There is an urgent need to comprehensively study the gut microbiota and resistome of individuals in these regions to better prepare for future public health security.

Previous investigations into the human resistome primarily concentrated on ARGs, often overlooking the complex factors that influence the evolution and spread of ARGs (15). Gut microbiota are known as ARG "reservoirs" and are closely related to the composition of ARGs (16). Compositional changes in microbial community structure are the main cause of changes in ARGs (17). In addition, when human pathogenic bacteria carrying virulence factor genes (VFGs) infect the host, antibiotic treatment is required, so there is a direct correlation between bacterial VFGs and ARGs (18). Moreover, the coexistence of ARGs and metal resistance genes (MRGs) has been reported to be common in human pathogens (19, 20). The concurrent selection of MRGs and ARGs may contribute to an increased AMR in bacteria (21, 22). Furthermore, the commensal bacteria in the human gut, hosting ARGs, can transfer ARGs to pathogens through mobile genetic elements (MGEs) (23–25), thereby increasing the risk of health hazards, including the emergence of multiple drug resistance (MDR) (26, 27). Hence, it is necessary to investigate the relationship between potential pathogens and ARGs, as well as the characteristics of ARGs, VFGs, MRGs, and MGEs within the genome of potential pathogens.

In this study, we conducted a comprehensive investigation based on gut metagenomes of 1,382 healthy Chinese individuals, including 415 individuals from the eastern populations (advanced industrialization) and 967 individuals from the western populations (developing industrialization). Using metagenomics and bioinformatics, we analyzed the microbial species profile, the ARG profile, and the genome of potential pathogens to gain insights into the gut microbiome and resistome of Chinese populations. Specifically, this study examined (1) the species and functional diversity of gut microbiota in eastern and western populations, (2) the characteristics of the ARG profile and core ARGs in eastern and western populations, and (3) the risk of potential pathogens in gut microbiota from eastern and western populations. Thus, this study advances our understanding of gut microbiome and resistome dynamics in eastern and western populations, offering vital implications for Chinese public health and poten-

tially aiding in the development of region-specific strategies for managing pathogenic infections.

## RESULTS

To investigate the characteristics of the gut microbiome and resistome across populations in eastern and western China, we conducted an integrative analysis on a large set of metagenomic data sets (28–32) (Table S1). One of the data sets, which was newly sequenced in the context of this study, comes from 70 Indigenous people in the Nagqu region of Tibet, representing people with traditional lifestyles in western China (Table S2; also see Materials and Methods). The collection comprises 1,382 metagenomic samples ($N_{eastern}$ = 415, $N_{western}$ = 967), along with curated host information, facilitating the assessment of gut microbiome characteristics in eastern populations (advanced industrialization) and western populations (developing industrialization) (Fig. 1A; Fig. S1; Table S3; also see Materials and Methods). The average age of the eastern populations is 36.54 years old, and the average age of the western populations is 34.34 years old, with no significant difference observed between the two groups (Table S3, Wilcoxon rank-sum test, $P$ = 0.11). Despite variations in size among the 6 data sets (Table S3; Fig. S2A), which include human metagenomes from 14 regions in China representing different host lifestyles, the integrated data set aids in characterizing the gut microbiome of populations in eastern and western China (Fig. S2B; also see Materials and Methods).

### Diversity landscapes in the microbiota of Chinese populations

The 1,382 Chinese human gut microbiome data were performed using the same quality control strategy (see Materials and Methods). To explore the characteristics of gut microbiota between eastern and western populations, we profiled the species and metabolic pathway compositions using MetaPhlAn 4 (v 4.1.0) (33) and HUMAnN 3 (v3.9) (34) (see Materials and Methods), respectively. Species accumulation curves demonstrated that with an increasing number of samples, the diversity of species observed in the gut microbiota of both eastern and western populations gradually approached a saturation point (Fig. 1B). The results indicate that our study has successfully captured most gut microbiota species within both the eastern and western populations, thereby offering robust data support for subsequent analyses. We further performed diversity analysis based on the species-level profiles of 1,382 samples. The findings revealed a substantial difference, indicating that the richness and Shannon index of the gut microbial species in western populations were significantly higher than those observed in eastern populations (Fig. 1C, Wilcoxon rank-sum test, ****$P$ < 0.0001). Considering that the human gut microbiome is associated with multiple factors, we conducted a stratified comparative analysis of populations from the eastern and western regions based on gender, ethnic groups, and lifestyles. The findings revealed that the richness and Shannon index of gut microbial species were significantly higher in western populations compared to eastern populations, regardless of gender, ethnic groups, and lifestyles (Fig. 1D through F, Wilcoxon rank-sum test, **$P$ < 0.01, ****$P$ < 0.0001). In addition, based on the microbial metabolic pathway profiles of 1,382 samples, we found that the richness and Shannon index of the gut microbial pathways in the western populations were significantly higher than those in the eastern populations (Fig. 2A, Wilcoxon rank-sum test, ****$P$ < 0.0001). Similarly, the comparison of microbial pathways between eastern and western populations, stratified by gender, ethnicity, and lifestyle, consistently indicated higher functional diversity among gut microbiota in the western populations (Fig. 2B through D, Wilcoxon rank-sum test, ****$P$ < 0.0001).

Next, we compared the diversity of gut microbial communities between eastern and western populations. We observed significant dissimilarity in the species composition of the gut microbial communities between the eastern and western populations (Fig. 2E), with inter-group differences notably greater than intra-group differences (Fig. 2E, ANOSIM: $R$ = 0.348, $P$ = 0.001; Fig. 2F, Wilcoxon rank-sum test, ****$P$ < 0.0001). Furthermore, the dissimilarity in species composition within the eastern populations was

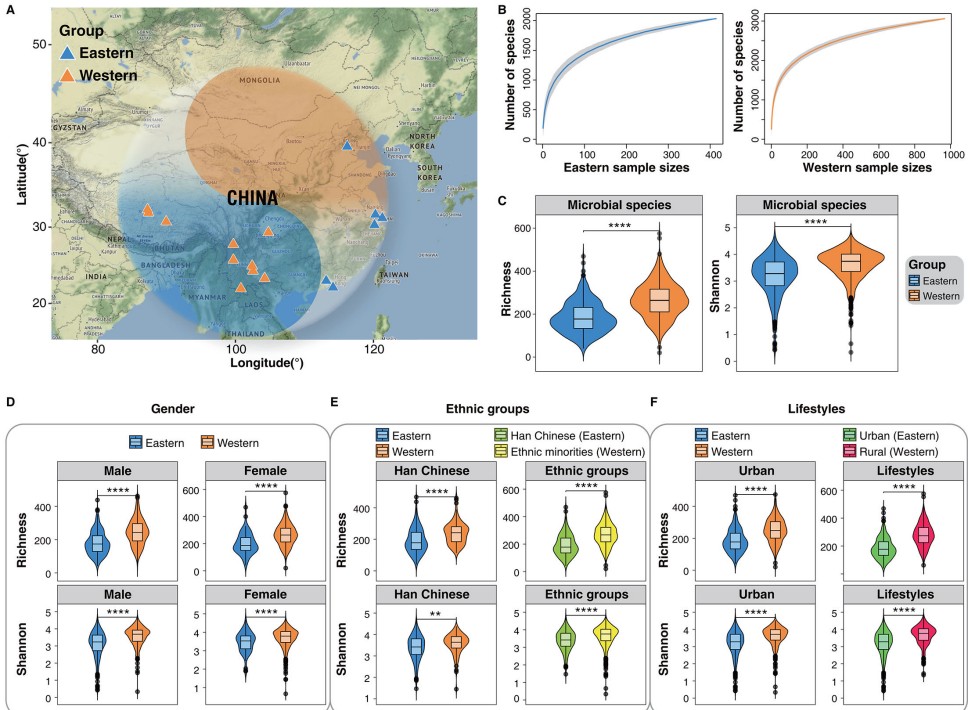

**FIG 1** Species diversity characteristics of gut microbiota in Chinese populations. (A) Geographical distribution of samples. The blue triangle signifies the geographic location of the eastern populations, while the orange triangle denotes the geographic location of the western populations. Refer to Table S3 for comprehensive details. (Map tiles by Stamen Design, under CC BY 4.0. Data by OpenStreetMap, under ODbL.) (B) Species cumulative curves of gut microbiota in eastern and western populations. The solid lines are the species cumulative curves (the mean number of species), while the light-colored areas represent their 95% confidence intervals. (C) Comparison of alpha diversity indices of gut microbiota between eastern and western populations. The violin plot shows information on richness and Shannon index between groups. Richness: Reflects the number of different species per sample. Higher richness suggests a more diverse sample in terms of different species. Shannon index: Represents a comprehensive measure of species richness and evenness in a sample. A higher Shannon index indicates greater richness and a more even distribution of species in the sample. (D) Comparisons of gut microbiota species richness and Shannon index were stratified by gender between eastern and western populations. (E) Comparisons of gut microbiota species richness and Shannon index were stratified by ethnic groups between eastern and western populations. The Han and minority ethnic groups in the western populations were compared separately with the Han ethnic group in the eastern populations. (F) Comparisons of gut microbiota species richness and Shannon index were stratified by lifestyles between eastern and western populations. The urban and rural residents in the western populations were compared separately with the urban residents in the eastern populations. The chosen significance level for the statistical test employed the Wilcoxon rank-sum test, **$P < 0.01$, ****$P < 0.0001$.

significantly higher than within the western populations (Fig. 2F, Wilcoxon rank-sum test, ****$P < 0.0001$). However, unlike the observed patterns in microbial species composition, we discovered greater overlap in the functional profiles of gut microbial communities between eastern and western populations (Fig. 2G). The results revealed statistically significant yet relatively minor disparities in the gut microbial metabolic pathways between eastern and western populations (Fig. 2G, ANOSIM: $R = 0.037$, $P = 0.007$). Although the functional profiles of the gut microbial communities are relatively similar between eastern and western populations, there are still significant differences in the relative abundance of 150 microbial pathways (Table S4, Wilcoxon rank-sum test, $P < 0.05$). Interestingly, the interindividual dissimilarity in gut microbial pathways within the eastern populations was not only significantly higher than that within the western populations but even significantly higher than that between eastern and western individuals (Fig. 2H, Wilcoxon rank-sum test, ****$P < 0.0001$).

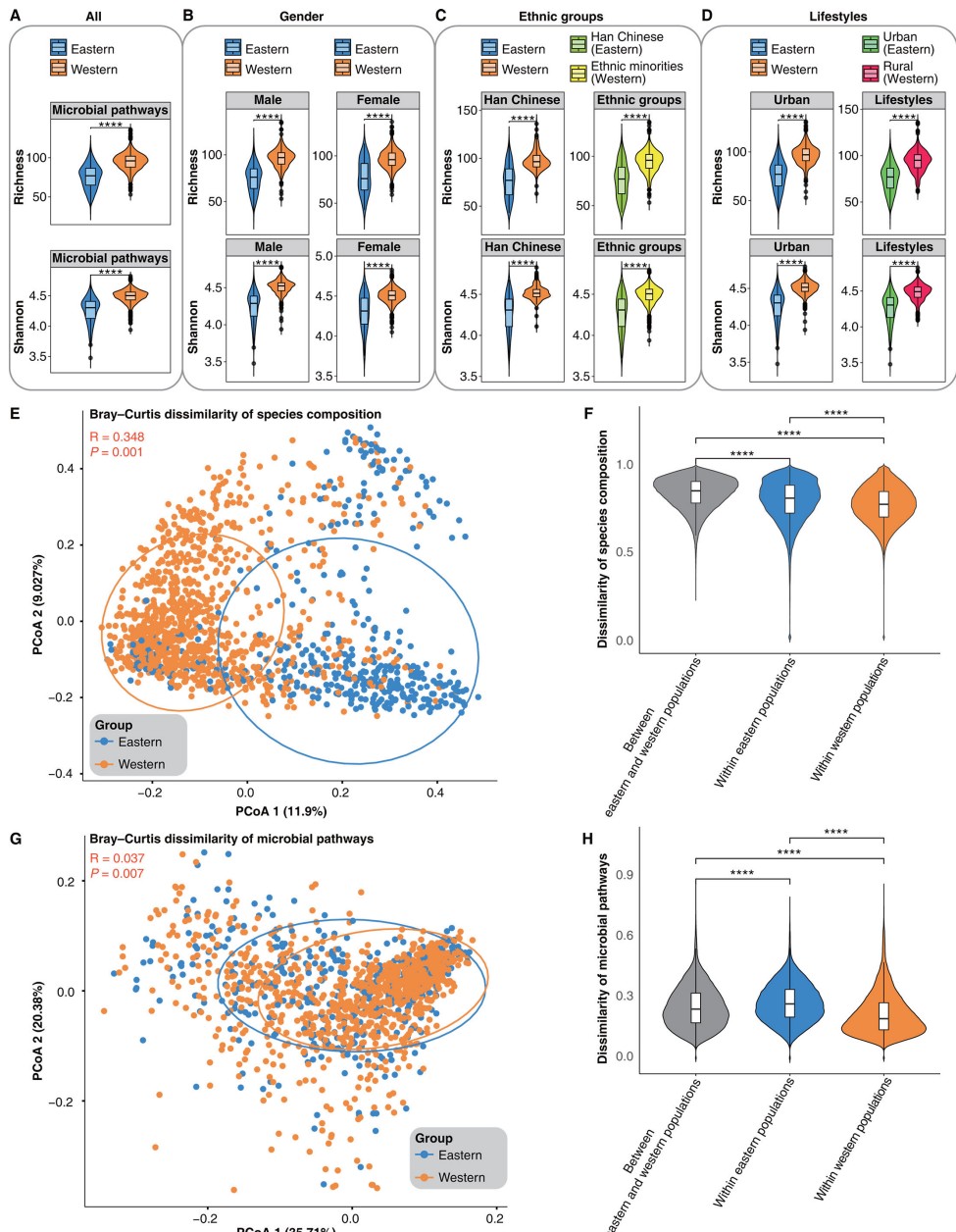

**FIG 2** Functional and community diversity characteristics of the gut microbiota in the Chinese populations. (A) Comparison of the richness and Shannon index of gut microbial pathways between eastern and western populations. (B) Comparisons of the richness and Shannon index of gut microbial pathways were stratified by gender between eastern and western populations. (C) Comparisons of the richness and Shannon index of gut microbial pathways were stratified by ethnic groups between eastern and western populations. The Han and minority ethnic groups in the western populations were compared separately with the Han ethnic group in the eastern populations. (D) Comparisons of the richness and Shannon index of gut microbial pathways were stratified by lifestyles between eastern and western populations. The urban and rural residents in the western populations were compared separately with the urban residents in the eastern populations. The comparisons are statistically significant (Wilcoxon rank-sum test, ****$P < 0.0001$). (E–H) Comparison of species and function profiles between eastern and western populations. The principal coordinate analysis (PCoA), utilizing the Bray-Curtis distance matrix, illustrates dissimilarity in the species composition (E) and microbial pathways (G) between eastern and western populations. The analysis of similarity (ANOSIM) was employed to statistically evaluate whether significant differences exist between the two groups. $R > 0$ indicates that dissimilarities between groups are greater than within groups. $R \approx 0$ indicates no difference between groups. $P < 0.05$ suggests that the observed differences between groups are statistically significant. Comparison of dissimilarity of

Fig 2 (Continued)

species composition (F) and microbial pathways (H) in gut microbial communities within the eastern populations, within the western populations, and between the eastern and western populations. The comparisons are statistically significant (Wilcoxon rank-sum test, ****$P < 0.0001$).

Overall, there are significant differences in the diversity of gut microbiota between eastern and western populations, but they may share a functional profile.

## Indicator microbial species features between eastern and western populations

To explore the taxonomic characteristics of gut microbiota between eastern and western populations, we further compared the composition of microbial communities. The results displayed a classification at the phylum level, spotlighting the top 10 microbial phyla by mean relative abundance in the gut microbiota of both eastern and western populations (Fig. 3A). The composition of the gut microbiota in both eastern and western populations showed a large variation among individuals (Fig. 3A). The mean relative abundance of Firmicutes was highest among the phylum-level compositions in the gut microbiota of both eastern and western populations (Table S5). We found significant differences in the mean relative abundance of 10 microbial phyla between the eastern and western populations (Fig. 3B). Specifically, the mean relative abundance of Bacteroidota, Proteobacteria, Candidatus_Melainabacteria, Lentisphaerae, and Synergistetes was higher in the gut microbiota of the eastern populations, while the mean relative abundance of Firmicutes, Actinobacteria, Euryarchaeota, Candidatus_Saccharibacteria, and Ascomycota was higher in the gut microbiota of the western populations (Fig. 3B; Table S5). Next, we performed indicator species analysis (see Materials and Methods), a widely employed method in ecological research (35, 36), which helps reveal key microbial species associated with specific populations. A total of 954 indicator species were identified, with 290 associated with the gut microbiota of eastern populations and 664 designated for the gut microbiota of western populations (Fig. 3C; Table S6). To further explore the characteristics of gut indicator species between the eastern and western populations, we performed a between-group enrichment analysis at the phylum level. The results showed that Bacteroidota, Proteobacteria, and Lentisphaerae were enriched in eastern populations, while Firmicutes, Actinobacteria, and Candidatus_Saccharibacteria were enriched in western populations (Fig. 3D, Fisher Exact Test, $P_{adj}$ <0.05). Importantly, according to the "Catalogue of Human Infectious Pathogens" (Enacted by the National Health Commission of the People's Republic of China in 2023) (Table S7), we found 72 indicator species that are potential pathogens related to human health (Fig. 3E, Table S6; also see see Materials and Methods). The potential pathogens are more enriched in gut microbial indicator species in western populations compared with eastern populations (Fisher Exact Test, $P_{adj}$ <0.0001). Next, we comprehensively investigated the distribution of potential pathogens in eastern and western populations at the species level based on the "Catalogue of Human Infectious Pathogens." A total of 218 potentially pathogenic microbial species were identified, each exhibiting a relative abundance of >0.001% in at least one sample. The mean relative abundance of 28 potential pathogens was significantly different in the gut microbiota between eastern and western populations, with the mean relative abundance of 25 potential pathogens being higher in the gut microbiota of the western populations (Table S7, Wilcoxon rank-sum test, $P < 0.05$). Moreover, we found that the richness of potential pathogens in the western populations is significantly higher than that in the eastern populations (Fig. 3F, Wilcoxon rank-sum test, ****$P < 0.0001$), regardless of gender, ethnic groups, and lifestyles (Fig. S3A, Wilcoxon rank-sum test, ****$P < 0.0001$).

In summary, we identified the characteristic gut microbiota between eastern and western populations. Notably, we observed significant differences in potential pathogen occurrence in the gut microbiota of these two populations.

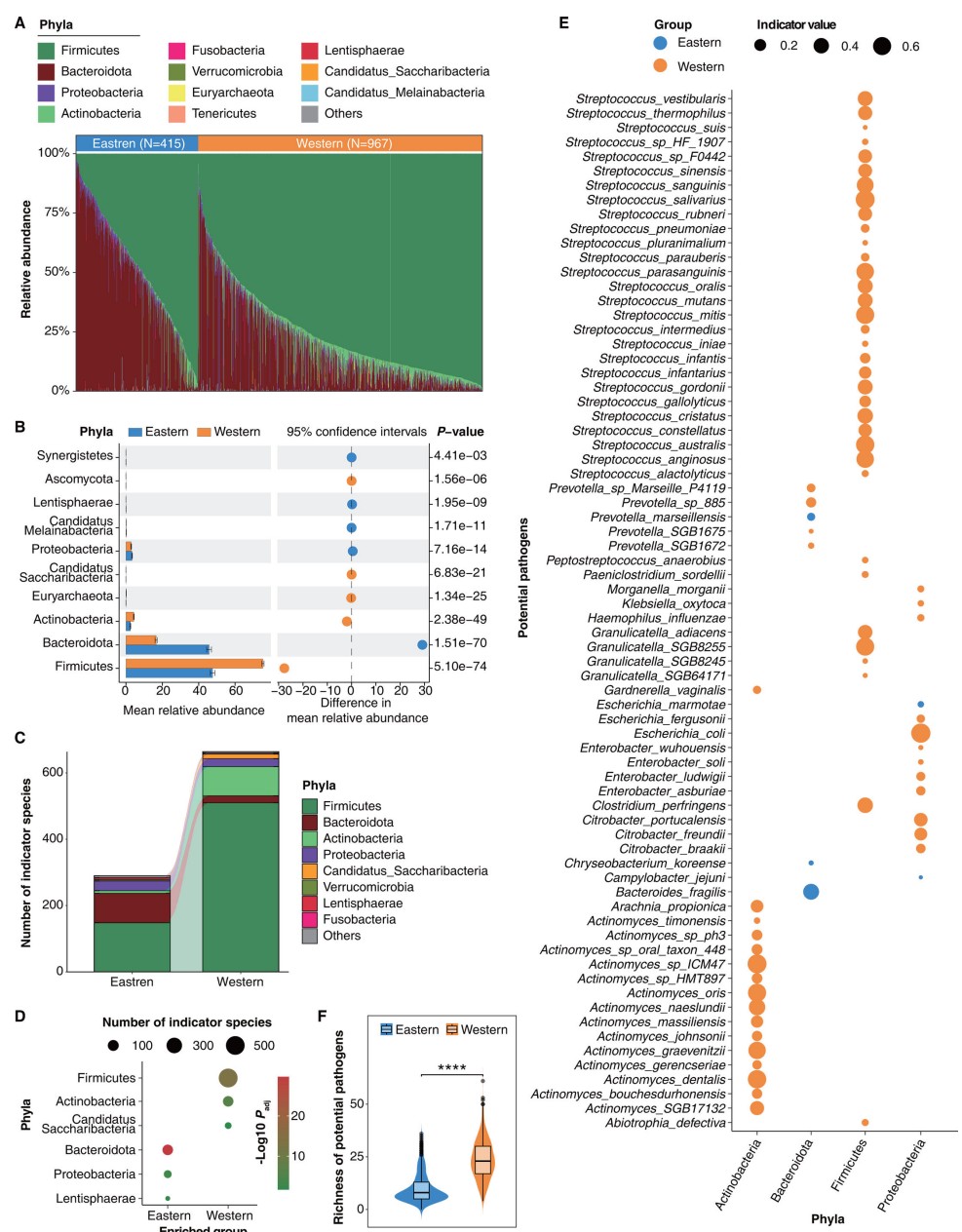

**FIG 3** Taxonomy characteristics of gut microbiota in eastern and western populations. (A) Variations in the composition of gut microbiota across eastern and western populations were plotted based on the mean relative abundance ranking of microbial phyla (top 10 for each group). (B) STAMP (Statistical Analysis of Metagenomic Profiles) analysis reveals significant differences in the mean relative abundance of gut microbiota at the phylum level between eastern and western populations. (C) Display of the number and phylum classification of gut indicator species in eastern and western populations. (D) Intergroup enrichment analysis of gut indicator species at the phylum level in eastern and western populations. The size of the circles represents the number of indicator species, and the color represents statistical significance. Statistical significance was verified through Fisher's exact test and multiple hypothesis testing corrections were performed using the false discovery rate (FDR) method. (E) Potential pathogens contained in gut indicator species of eastern and western populations (Table S6 and Materials and Methods). The size of the circle represents the indicator value, the blue dots are the gut indicator species for the eastern population, and the orange dots are the gut indicator species for the western population. (F) Comparison of the richness of potential pathogens in the gut microbiota between eastern and western populations (Wilcoxon rank-sum test, ****$P$ < 0.0001).

## ARG profiles across Chinese gut microbiomes

Considering that antibiotic treatment is often necessary following pathogenic bacterial infections (18), we hypothesize a direct correlation between potential pathogens and ARGs. Next, we employed the integrated analysis pipeline of ARGs-OAP (v3.2.4) (37) to accurately delineate the profiles of ARGs within 1,382 Chinese gut metagenomic samples (see Materials and Methods). A total of 28 ARG types were identified (Table S8). In the gut microbiota of the eastern populations, the top three ARG types based on mean relative abundance are tetracycline, macrolide-lincosamide-streptogramin, and beta-lactam, while in the western populations, they are tetracycline, macrolide-lincosamide-streptogramin, and multidrug (Table S8). In contrast to the distribution pattern observed at the phylum level in gut microbiota (Fig. 3A), there is relatively little variation in gut ARG types between individuals in the eastern and western populations (Fig. 4A). We found that the relative abundance of ARGs in the gut microbiota of western populations was significantly higher than those in eastern populations (Fig. 4B, Wilcoxon rank-sum test, $P = 8.30e{-}06$), regardless of gender, ethnic groups, and lifestyles (Fig. S3B, Wilcoxon rank-sum test, $*P < 0.05$, $***P < 0.001$, $****P < 0.0001$). To further explore the characteristics of gut ARG types, we computed the core ARG types separately for the eastern and western populations (see Materials and Methods). The core ARGs were defined as those that could be identified in 90% of the samples in each group (38). Interestingly, we found that these two populations shared the same core ARG types (Table S8). The mean relative abundance of macrolide-lincosamide-streptogramin and mupirocin core ARG types did not exhibit significant differences between the eastern and western populations (Table S8). In addition, the mean relative abundance of beta-lactam and chloramphenicol core ARG types was notably higher in the eastern populations compared to the western populations (Fig. 4C; Table S8, Wilcoxon rank-sum test, $P < 0.05$). Conversely, the mean relative abundance of core ARG types, including trimethoprim, vancomycin, bacitracin, tetracycline, aminoglycoside, pleuromutilin tiamulin, polymyxin, other peptide antibiotics, sulfonamide, and multidrug, was significantly higher in the western populations than in the eastern populations (Fig. 4C; Table S8, Wilcoxon rank-sum test, $P < 0.05$).

Next, 1,776 ARG subtypes were identified for a high-resolution examination of the characteristics of the gut resistome, revealing significant differences in the relative abundance of 894 ARG subtypes between the eastern and western populations (Table S9, Wilcoxon rank-sum test, $P < 0.05$). We further identified core gut ARG subtypes separately for the eastern and western populations (see Materials and Methods). The results showed that the eastern populations had 68 core ARG subtypes, with $tet(Q)$, $erm(F)$, and $erm(B)$ being the top 3 in terms of mean relative abundance, whereas the western populations had 92 core ARG subtypes, with $erm(B)$, $tet(Q)$, and $tet(O)$ ranking as the top 3 (Table S9). The eastern and western populations shared 64 core ARG subtypes, among which 43 core ARG subtypes had significantly higher relative abundances in the gut microbiota of the western populations, while 14 core ARG subtypes had significantly higher relative abundances in the gut microbiota of the eastern populations (Fig. 4D; Table S9). The western populations had more unique core ARG subtypes compared to the eastern populations (Fig. 4D). Moreover, the diversity analysis results revealed that the richness of gut ARG subtypes in western populations is significantly higher than that in eastern populations (Fig. 4E; Fig. S3C, Wilcoxon rank-sum test, $***P < 0.001$, $****P < 0.0001$).

Interestingly, we found that increases in GDP per capita may lead to a decrease in the richness of gut potential pathogens (Fig. 4F, $R = -0.47$, $P = 1.61e{-}77$) and ARG subtypes (Fig. 4G, $R = -0.30$, $P = 9.47e{-}30$), implying that the level of industrialization may influence the gut resistome. Finally, we explored the correlation between potential pathogens and ARGs, observing a strong positive correlation between the richness of potential pathogens and the richness of ARG subtypes (Fig. 4H, $R = 0.57$, $P = 4.45e{-}120$).

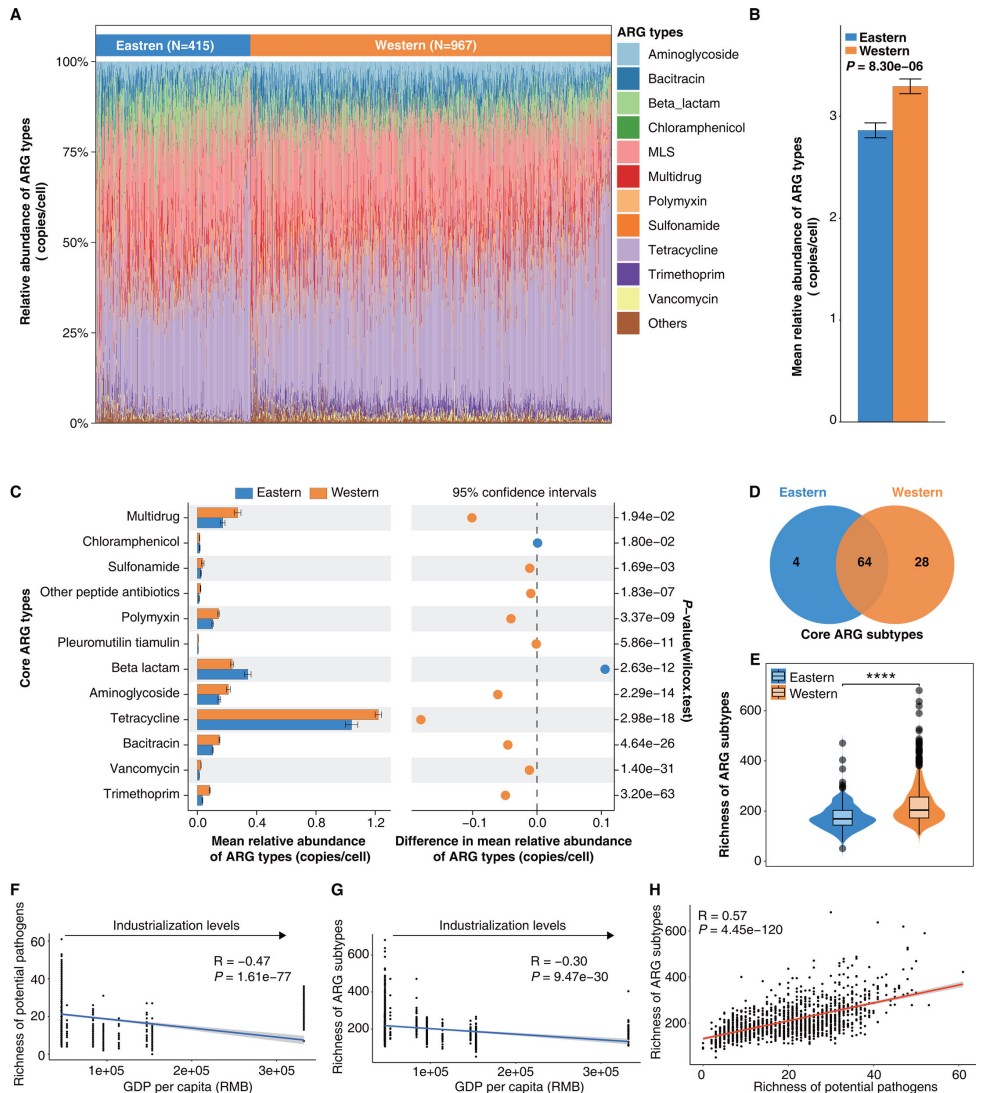

**FIG 4** Abundance and richness of gut ARGs in eastern and western populations. (A) Variations in the composition of gut ARG across eastern and western populations were plotted based on the mean relative abundance (copies/cell) ranking of ARG types (top 10 for each group). (B) Comparison of mean relative abundance (copies/cell) of gut ARG types in eastern and western populations (Wilcoxon rank-sum test, $P = 8.30e{-}06$). (C) STAMP analysis reveals significant differences in the mean relative abundance (copies/cell) of gut core ARG types between eastern and western populations. (D) Venn diagram of gut core ARG subtypes in eastern and western populations. (E) The violin plot demonstrates significant differences in the richness of gut ARG subtypes between eastern and western populations (Wilcoxon rank-sum test, ****$P < 0.0001$). As industrialization levels increase, the richness of gut potential pathogens (F) and ARG subtypes (G) in Chinese populations decreases. In a broad sense, high GDP per capita indicates a high level of industrialization (39). (H) The richness of gut potential pathogens in the Chinese populations is positively correlated with the richness of gut ARG subtypes. Spearman's rank correlation coefficient ($R$) and statistical significance ($P$) are calculated. Each dot represents a sample, and light gray areas represent their 95% confidence intervals.

## Mining potentially pathogenic microbial genomes

Using *de novo* metagenome assembly, we reconstructed a total of 30,547 high-medium quality metagenomic assembly genomes (MAGs), representing 1,937 species-level genome bins (SGBs), to further explore potential pathogens in the gut microbiota of the Chinese populations (see Materials and Methods). These genomes were filtered with a quality threshold of completeness ≥50% and contamination ≤5%, coupled with

a genome quality score (QS; calculated as completeness − 5 × contamination) ≥ 50% (Table S10). In all, 15,120 MAGs (accounting for 49.5% of the total MAGs) were >90% complete and <5% contaminated, hereafter referred to as "near-complete" genomes (Fig. 5A; Table S10). A subset of 2,002 MAGs (13.2% of near-complete genomes) had 5S, 16S, and 23S rRNA genes as well as at least 18 of the standard tRNAs, which can be classified as the "high-quality" draft genomes based on the MIMAG standard (40). The rest of the catalog consists of 15,427 medium-quality MAGs (≥50% completeness and ≤5% contamination) (Fig. 5A; Table S10).

We annotated the 1,937 SGBs using the Genome Taxonomy Database Toolkit (GTDB-Tk) (v2.3.2, reference database version r214) (42). Based on taxonomic information, a total of 91 SGBs (1,127 MAGs) are listed in the "Catalogue of Human Infectious Pathogens," representing potential pathogens (Table S11; also see Materials and Methods). We then proceeded to perform functional annotation on the 1,127 potentially pathogenic microbial genomes, covering ARGs, VFGs, MRGs, and MGEs (see Materials and Methods). The definition of potential pathogens does not take into account whether the microbial genome carries VFGs. Due to strain heterogeneity, the genomes of potential pathogens in this study may not contain VFGs (Table S11). The results revealed that 27.2% (307/1,127) of MAGs harbored ARGs, 35.3% (398/1,127) harbored VFGs, 32.2% (363/1,127) harbored MRGs, and 98.9% (1115/1,127) harbored MGEs (Table S11; Fig. 5B). It is worth noting that 22.2% (250/1,127) MAGs carrying both ARGs and VFGs were defined as potentially pathogenic antibiotic-resistant bacteria (PARB) (41) (Fig. 5B). PARB play a crucial role in evaluating health risks associated with antibiotic resistance due to their capability to induce diseases and undermine the effectiveness of antibiotic treatments (41). Meanwhile, we found that 98.4% (246/250) of PARB carried MRGs and MGEs, which also suggested the close connection between ARGs, VFGs, MRGs, and MGEs in the genomes of potential pathogens (Fig. 5B). We further investigated the relationships between ARGs and VFGs, ARGs and MRGs, and ARGs and MGEs in the genomes of potential pathogens. To avoid the influence of phylogenetic inertia, we employed the method of phylogenetic independent contrasts (PIC) to calculate correlations. The results consistently showed that there is a significant positive correlation between the number of ARGs and the number of VFGs, MRGs, and MGEs in the genome of potential pathogens (Fig. S4A through C).

Next, we estimated the risk of PARB in the eastern and western populations by analyzing its richness and relative abundance (see Materials and Methods). The results showed that the richness of gut PARB in the western populations was significantly higher than that in the eastern populations (Fig. 5C, Wilcoxon rank-sum test, $P = 6.27e-51$). The mean relative abundance of 66 PARB was significantly higher in the eastern populations, primarily comprising strains from the genera *Klebsiella* and *Enterobacter* (Fig. 5D; Table S12). By contrast, the mean relative abundance of 173 PARB was higher in the western populations, with strains from the genus *Escherichia* being predominant (Fig. 5D; Table S12). Overall, the western populations have a greater diversity of PARB in their gut microbiota compared to the eastern populations.

Finally, we characterized the number of ARGs, VFGs, MRGs, and MGEs in genomes of potential pathogens at the strain level. The results showed that some potentially pathogenic microbial species contained multiple genomes, and there was large variation in the number of ARGs, VFGs, MRGs, and MGEs in different strains (Fig. 5E). By visual inspection, we found many ARGs, VFGs, MRGs, and MGEs in the genomes of Enterobacteriaceae species, which suggests that potential pathogens of Enterobacteriaceae deserve special attention (Fig. 5E). We further performed redundancy analysis on ARGs, VFGs, MRGs, and MGEs in multiple genomes contained in each SGB (see Materials and Methods). The results showed that the Enterobacteriaceae SGB_1871 (*Escherichia coli*) had the largest number of non-redundant ARGs, VFGs, MRGs, and MGEs (Fig. S5). The *E. coli* carrying VFGs and many ARGs may pose a major threat to human health. For example, an *E. coli* epidemic broke out in Germany in 2011, causing acute diarrhea,

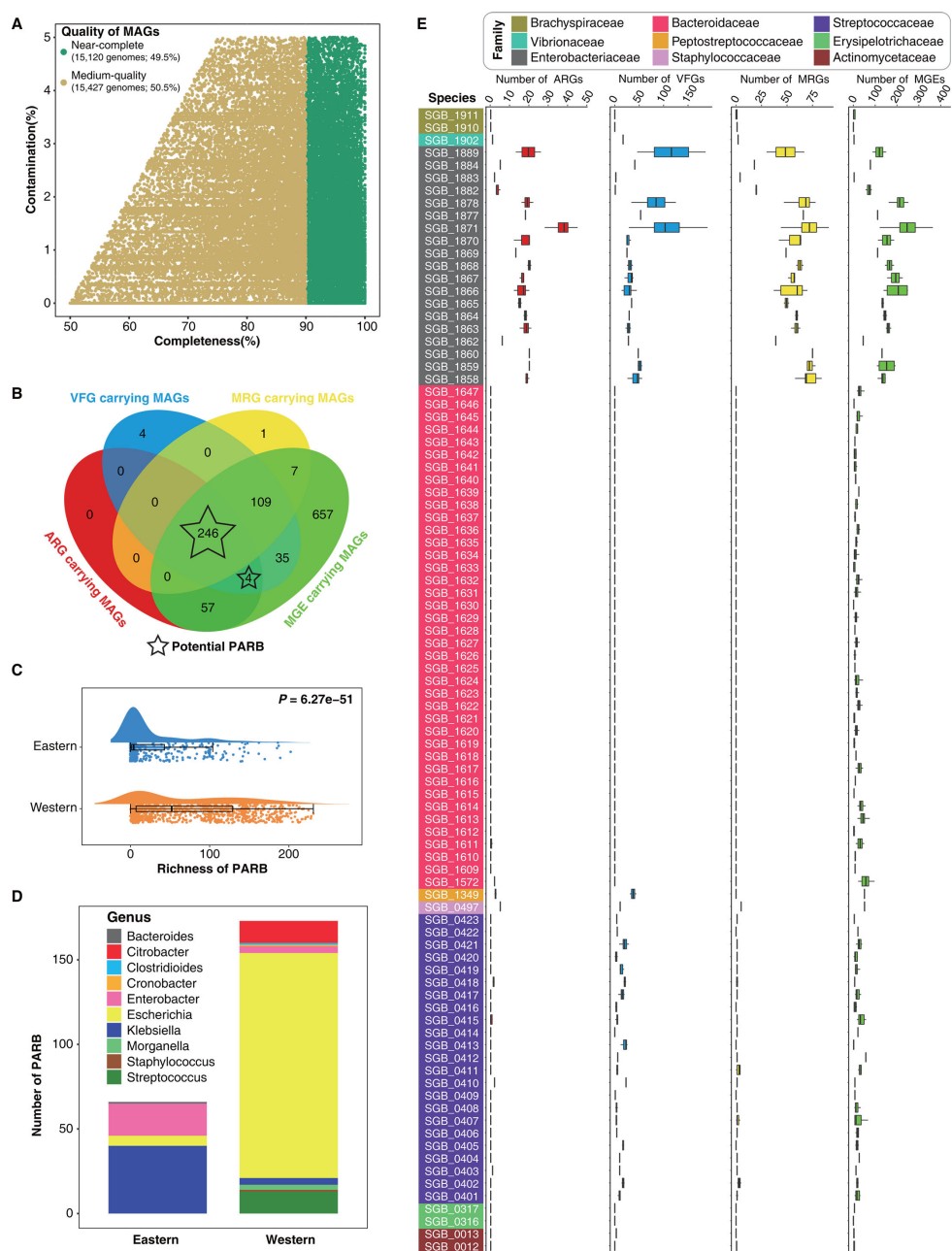

**FIG 5** Overview of ARGs, VFGs, MRGs, and MGEs in the genome of gut potential pathogens. (A) Completeness and contamination scores for each of 30,547 MAGs. (B) Statistics of the number of ARGs, VFGs, MRGs, and MGEs carried by MAGs belonging to potential pathogens assembled in this study. MAGs carrying both ARGs and VFGs were defined as potentially pathogenic antibiotic-resistant bacteria (PARB) (41). (C) The richness of PARB in the human gut microbiota in western China is significantly higher than that in eastern China (Wilcoxon rank-sum test, $P = 6.27e-51$). (D) The number and bacterial genus classification of PARB with significantly higher mean relative abundance in the gut microbiota of the eastern or western populations (Table S12). (E) Statistics of the number of ARGs, VFGs, MRG, and MGEs in the strain-level genomes of 91 potentially pathogenic microbial species assembled in this study.

abdominal pain, and even death in humans (43). The culprit responsible for the outbreak is a strain of *E. coli O104:H4* that carries Shiga toxin 2 (*stx2*) and many ARGs (43).

## Dynamics of *E. coli* populations in Chinese gut microbiomes

To assess the potential risks of *E. coli* in Chinese gut microbiota, we downloaded the reference genomes of the *O104:H4* strain and performed phylogenetic analysis and average nucleotide identity identification with the 90 near-complete *E. coli* genomes assembled in this study ($N_{eastern}$ = 47, $N_{western}$ = 43). The phylogenetic tree results showed that *O104:H4* has the closest genetic relationship with the *E. coli* genome (MAG_22164) assembled from the human gut microbiomes in eastern populations (BJ) and the average nucleotide identity exceeds 98% (Fig. 6A; Table S13). We further searched for VFGs in all *E. coli* genomes assembled in this study, and fortunately, no VFGs encoding Shiga toxin were found. Moreover, we observed that a mean ARG count of 48.7, a mean VFG count of 201.4, a mean MRG count of 107.9, and a mean MGE count of 636.1 carried by the *E. coli O104:H4* (Fig. 6A; Table S14). The number of ARGs, VFGs, MRGs, and MGEs in *E. coli* O104:H4 is significantly higher than those found in the gut *E. coli* of both eastern and western populations (Fig. 6B, Wilcoxon rank-sum test, ****$P$ < 0.0001). This suggests that potential pathogens with a large number of ARGs, VFGs, MRGs, and MGEs in their genomes need to be closely monitored.

We next compared the characteristics of the gut *E. coli* genomes between eastern and western populations. Multidrug ARGs, motility-related VFGs, multi-metal resistance genes, and MGEs for replication/recombination/repair were found to be the most abundant in the genomes of gut *E. coli* in both eastern and western populations (Fig. S6A through D; Table S15). The results showed that the number of ARGs and MRGs is not significantly different in the *E. coli* genomes between the eastern and western populations (Fig. 6B). However, the number of aminocoumarin and aminoglycoside ARGs in genomes of gut *E. coli* in the eastern populations was significantly higher than that in the western populations (Fig. S6A, Wilcoxon rank-sum test, *$P$ < 0.05, **$P$ < 0.01). Remarkably, the number of VFGs and MGEs in the gut *E. coli* genomes of western populations is significantly higher than those in eastern populations (Fig. 6B, Wilcoxon rank-sum test, *$P$ < 0.05, ***$P$ < 0.001). Specifically, the VFGs in the types of effector delivery system, adherence, antimicrobial activity/competitive advantage, nutritional/metabolic factor, invasion, and motility in gut *E. coli* in the western populations were significantly higher than those in the eastern populations (Fig. S6B, Wilcoxon rank-sum test, *$P$ < 0.05, **$P$ < 0.01, ****$P$ < 0.0001). In the types of MGEs, those associated with integration/excision and stability/transfer/defense are more prevalent in the gut *E. coli* of the western populations (Fig. S6D, Wilcoxon rank-sum test, *$P$ < 0.05). Conversely, MGEs linked to replication/recombination/repair exhibit a higher count in the gut *E. coli* of the eastern populations (Fig. S6D, Wilcoxon rank-sum test, ***$P$ < 0.001). Moreover, there was no significant difference in the number of phages and transfer MGEs in the gut *E. coli* genomes between eastern and western populations (Fig. S6D).

Overall, comparative genomics analysis revealed that human gut *E. coli* harbors a substantial number of resistome-related genes, potentially elevating health risks. Notably, significant differences were observed in the resistome profiles of *E. coli* between eastern and western populations.

## DISCUSSION

This study provides the first comprehensive overview of the human gut microbiota between eastern and western China at the metagenomic level. We found that compared with western populations, the interindividual dissimilarity of microbial species composition increased significantly in eastern populations, while the intraindividual species diversity (richness and Shannon index) decreased significantly. These changes are consistent with previously reported effects of industrialization on the gut microbiota (44–46). Moreover, the mean relative abundance of Bacteroidota in the gut microbiota of the eastern populations is 2.75 times that of the western populations. Reportedly, the Bacteroidota are enriched in the industrial samples compared to the palaeofaeces and the non-industrial samples (47). Given the relatively high level of industrialization and urbanization in the eastern region of China, accompanied by a gradual westernization

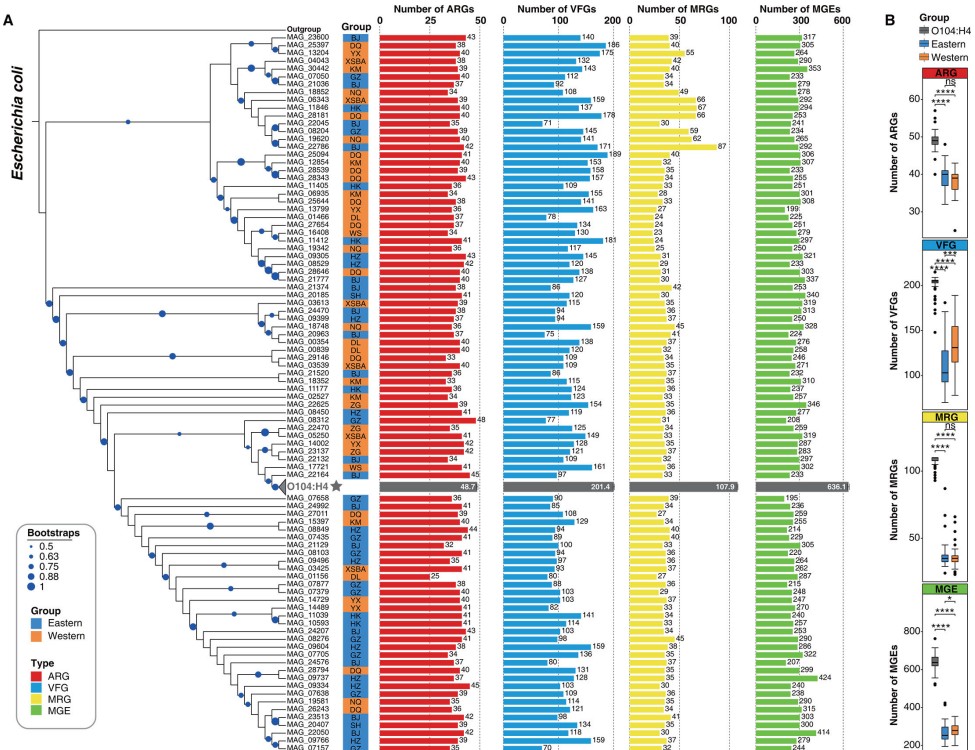

**FIG 6** Differences in ARGs, VFGs, MRGs, and MGEs of gut *E. coli* between eastern and western populations. (A) The phylogenetic tree shows the phylogenetic relationships of 47 near-complete *E. coli* genomes from the gut microbiota of eastern populations and 43 near-complete *E. coli* genomes from the gut microbiota of western populations. A total of 60 *E. coli* O104:H4 strains were obtained from public databases, see Table S14 for details. The blue dots above the phylogenetic tree represent bootstraps values. The bar plot on the right represents the number of ARG, VFG, MRG, and MGE carried by the corresponding *E. coli* genome, respectively. The eastern populations include Beijing (BJ), Guangzhou (GZ), Hong Kong (HK), Hangzhou (HZ), and Shanghai (SH). The western populations include Yuxi (YX), Diqing (DQ), Xishuangbanna (XSBA), Kunming (KM), Nagqu (NQ), Dali (DL), Wenshan (WS), and Zigong (ZG). (B) The genomic analysis reveals distinct profiles of ARGs, VFGs, MRGs, and MGEs in *E. coli* O104:H4, *E. coli* from the eastern populations, and those from the western populations. Wilcoxon rank-sum test was employed for inter-group comparisons, *P < 0.05, ****P < 0.0001.

of dietary habits (48), we speculate that the gut microbiota of eastern populations has shifted to better align with the metabolic requirements of the host (1). Despite the impact of industrialization on altering the gut microbiota composition between eastern and western populations, the functional characteristics of gut microbial communities remain similar, which reflects the functional redundancy in the human microbiome (49). It is worth noting that interindividual dissimilarity in microbial metabolic pathways is heightened in the eastern populations. This may be due to more frequent horizontal gene transfer (HGT) in the microbiomes of industrialized populations (50), leading to functional divergence, although further evidence is needed to confirm this.

We recovered ARG profiles representing the Chinese human gut resistome from 1,382 gut metagenomes and identified core ARGs in both eastern and western populations, respectively. Our analysis revealed that, compared to the eastern populations, the western populations had a significant increase in both the abundance and diversity of ARGs within their gut microbiota, along with a higher prevalence of unique core ARG subtypes. The findings are consistent with a recent sewage surveillance-based study (51), which concluded that sewage from the Tibetan Plateau exhibited higher ARG richness and abundance compared to that of eastern China (51). However, previous studies have confirmed that the burden of ARGs in the environment of eastern China is significantly higher than that in western China (10–12). This phenomenon indicates

that environmental factors are only one of the driving forces behind gut resistome in Chinese populations. We found that the level of industrialization, measured by GDP per capita, may influence gut resistome in Chinese populations. The richness of potential pathogens in the gut microbiota is significantly higher in western populations compared to eastern populations. Furthermore, there is a significant positive correlation between the richness of potential pathogens and the richness of ARG subtypes. We speculate that the more abundant potential pathogens in the western population may be directly related to higher levels of gut ARGs.

In addition, we found that the gut microbiota of the eastern and western populations contained the same core ARG types and shared most of the core ARG subtypes, suggesting that people may have common preferences for antibiotics (10). Importantly, we found that the relative abundance of two core ARG types is significantly higher in the eastern populations, while the relative abundance of eight core ARG types is significantly higher in the western populations. This may indicate higher consumption of specific antibiotics in different regions. Based on the prior investigation, a significant positive correlation has been identified between the abundance of beta-lactam class ARGs in the human gut microbiota and the consumption of beta-lactam antibiotics (52). We found that the mean relative abundance of beta-lactam ARGs in the gut microbiota of the eastern populations was approximately 145% of that in the western populations. This suggests that the consumption of beta-lactam antibiotics in the eastern populations may be higher than that in the western populations.

We further utilized metagenomic assembly to obtain 91 potentially pathogenic microbial species, representing 1,127 genomes of potential pathogens. Notably, a total of 250 PARB strains were identified in the gut microbiota of the Chinese populations. Our findings suggest that the western populations in China may face a higher potential risk from PARB, particularly from strains of the genus *Escherichia*. By contrast, the eastern populations should focus on monitoring strains from the genera *Klebsiella* and *Enterobacter*.

Moreover, we found that the potentially pathogenic genomes of Enterobacteriaceae contained a substantial number of ARGs, VFGs, MRGs, and MGEs. Furthermore, we observed that *E. coli*, a member of the Enterobacteriaceae family, harbors the highest number of non-redundant ARGs, VFGs, MRGs, and MGEs. Although there was no significant difference in the number of ARGs and MRGs within gut *E. coli* between the eastern and western populations, a notable distinction emerged in the number of MGEs and VFGs. The western populations exhibited a significantly higher count of MGEs and VFGs in gut *E. coli* compared to the eastern populations, suggesting an elevated risk of *E. coli* pathogenicity in western populations. The large number of VFGs and MGEs in the *E. coli* genome may increase the level of risk to human health. For example, *E. coli* O104:H4 may have acquired VFGs encoding *stx2* through the MGEs (53–55). In our comparative genomic analysis with the highly pathogenic strain O104:H4, we observed a substantial reduction in the number of ARGs, VFGs, MRGs, and MGEs in the gut *E. coli* genome of healthy Chinese individuals. This implies a potential collaborative role of ARGs, VFGs, MRGs, and MGEs in enhancing the pathogenicity of *E. coli*.

In total, our study has unveiled that industrialization may have potentially induced unexpected alterations in the gut microbiome and resistome between eastern and western populations. These findings hold significant implications for Chinese public health, offering insights that could facilitate the development of region-specific strategies for managing pathogenic infections. Under the guidance of the One Health approach, data from the Chinese gut microbiomes can provide support for understanding the spread of ARGs in the environment, animals, and humans. The human gut microbiome is associated with multiple factors, and further research and data are still needed to support our conclusions.

## MATERIALS AND METHODS

### Chinese metagenomic data sets

To comprehensively explore the characteristics of the Chinese gut microbiome and resistome, we selected 1,383 healthy Chinese fecal samples from 6 metagenomic data sets (28–32). A total of 1,382 metagenomes with ≥3 million reads were included in the analysis, consisting of 415 samples from eastern populations and 967 samples from western populations (Table S3), with metagenomes lacking sufficient sequencing depth being excluded. These data sets were accompanied by metadata, facilitating the assessment of microbiome and resistome disparities between eastern and western populations (Table S3). In this study, the term "healthy population" refers to a cohort of individuals characterized by physical well-being, typically devoid of significant illness or disease, and who have not received antibiotic treatment in the past month. To identify potential batch effects, we use UMAP to plot samples in these data sets, there is a lot of overlap among different studies (Fig. S2A). When we regrouped according to eastern and western populations (Fig. S2B), the major dissimilarity trends we described were recapitulated (Fig. 2E). These results indicated that the batch effect observed in this study may have been minimal.

### Newly sequenced metagenomic data sets

We collected 70 healthy human fecal samples from three villages in Nagqu, Tibet Autonomous Region, in July 2020. Using the MGIEasy fecal sample collection kit (Cat. 10000035265, BGI, China), the inner parts of the fresh fecal samples of all 70 individuals were collected with a sterile spatula and then kept in the sterile sampling tubes. All fecal samples were stored in −80°C freezers until transported to China National Gene Bank (CNGB), Shenzhen on dry ice for further sequencing analysis. DNA extraction of 70 fresh human fecal samples was performed using MagPure Stool DNA KF Kit B (Cat. MD5115, Magen) according to the standard kit protocol. DNA sequencing libraries were constructed by the MGIEasy Universal DNA Library Prep Set (Cat. 1000006986) and then sequenced on the DNBSEQ-T1 machine for $2 \times 150$ bp paired-end reads. After DNA extraction and metagenomics sequencing, over 1,800 GB of raw data (mean ± SD = 26.45 ±10.45 GB per sample) was generated from 70 samples, with an average coverage of 97.21% ± 1.34% (Table S2). The detailed information for the samples is shown in Table S2.

### Data set classification

According to the socioeconomic development status of China (56), we divided the metagenomic samples into two groups: eastern populations and western populations (Fig. 1A; Table S3). In a broad sense, industrialization is a process characterized by sustained increase in per capita income (39). Based on the GDP per capita of various provinces and regions in China over the past decade (data source: National Bureau of Statistics of China; https://data.stats.gov.cn/), the eastern region represented in this study is defined as having advanced industrialization, as it ranks among the top 10 in the country (Fig. S1). By contrast, the western region is defined as having developing industrialization due to its relatively lower GDP per capita (Fig. S1). The 415 samples in eastern China mainly belong to the urbanized Han ethnic group, including Beijing ($n = 150$), Hangzhou ($n = 114$), Hongkong ($n = 54$), Guangzhou ($n = 54$), Shanghai ($n = 24$), and Wuxi ($n = 19$). The 967 samples collected from western China encompass a diverse range of demographics, including urban and rural residents, as well as individuals from both the Han ethnic group and various indigenous ethnic groups. For instance, in Kunming, the samples are drawn from six ethnic groups: Han ($n = 64$, rural residents), Han ($n = 44$, urban residents), Zang ($n = 82$, urban residents), Miao ($n = 79$, urban residents), Bai ($n = 83$, urban residents), Dai ($n = 84$, urban residents), and Hani ($n = 82$, urban residents). In addition, other locations like Diqing ($n = 118$, ethnic Zang and rural residents), Xishuangbanna ($n = 81$, ethnic Dai and rural residents), Nagqu ($n = 70$, ethnic

Zang and rural residents), Yuxi ($n$ = 56, ethnic Hani and rural residents), Dali ($n$ = 55, ethnic Bai and rural residents), Wenshan ($n$ = 53, ethnic Miao and rural residents), and Zigong ($n$ = 16) are included in the study, each contributing valuable insights into the diverse ethnic and geographic landscape of western China. The study covered a wide range of regions, each with characteristics of either eastern or western populations. In this study, the term "minority ethnic groups" refers to five indigenous ethnic groups in China, namely Zang (Tibetan), Dai, Hani, Bai, and Miao. Using small effect sizes (f = 0.1), we performed power analyses for this study with a significant level of 0.05. The results revealed a statistical power surpassing 0.85, indicating a robust statistical foundation for the study's findings.

## Metagenomic quality filtering

The raw reads of 1,382 metagenomic samples underwent adapter trimming and quality filtering using fastp (v0.23.0) (57) with default parameters. Human genomes (RefSeq assembly accession: GCF_000001405.40) were used to build the host genomes index, and host contamination reads were removed by the Bowtie 2 (v2.5.1) (58) alignments, yielding clean metagenomic reads to be used for succeeding analyses.

## Metagenomic processing

Microbial community taxonomic profiling was performed on all the 1,382 samples with MetaPhlAn 4 (v 4.1.0) (33) with default parameters based on the mpa_vJun23_CHOC-OPhlAnSGB_202403 database. An annotated species with relative abundances lower than 0.001% was removed from the taxonomic profiles. To assess the diversity of gut microbiota between the eastern and western populations, we used the R vegan (v2.6.4) package to calculate the alpha diversity (both richness and Shannon) and beta diversity (based on Bray-Curtis dissimilarity) of 1,382 samples. The PCoA was performed with the ade4 (v1.7.22) package in R using the Bray-Curtis distance matrix among samples. Analysis of similarities (ANOSIM, 999 permutations) was performed using the R vegan (v2.6.4) package. Assessment of the encoded microbial pathways was conducted using HUMAnN 3 (v3.9) (34). We filter out pathways with less than 50% coverage in each sample. We normalized the abundance of the remaining pathways to relative abundance using an additional script provided by HUMAnN3 (v3.9) (34). The statistically significant indicator species between eastern and western populations were detected using the function Indval in the R package labdsv (v 2.1.0) (36).

## ARG annotation and quantification

ARG identification of clean metagenomic reads was conducted using the latest version of ARGs-OAP (v3.2.4), employing recommended parameters. These included a similar cut-off of 80%, a query length coverage ratio of 75%, and an e-value of 1e-7, ensuring both high precision and sensitivity. The identified ARGs were annotated at the type/subtype level using the structured ARG (SARG) database. The relative abundance of ARGs was quantified using a universal unit of copies of ARG per cell, which was normalized against the cell counts in the analyzed metagenome. This normalization method, based on the identification of essential single-copy marker genes, is widely recognized and accepted (59). The core ARGs were defined as those that were present in at least 90% of the samples in each group (38), identified using the "compute_core_microbiome.py" script supplied in QIIME1 (60).

## Detection of potential pathogens

The potential pathogens identified in this study represent microbial species that either cause or have the potential to cause disease in humans. We determined specific potential pathogens (bacteria and fungi) information based on the latest revision of the "Catalogue of Human Infectious Pathogens" Enacted by the National Health Commission of the People's Republic of China in 2023 (Index Number: 000013610/2023–00666). The

"Catalogue" draws on relevant international and domestic regulations and research findings to scientifically assess the infectiousness of pathogenic microorganisms, the severity of the harm they can cause to individuals or groups upon infection, and China's capabilities and progress in infectious disease prevention and treatment. In addition, it takes into full consideration the practical needs of research, detection, and diagnosis of pathogenic microorganisms. To detect the existence of potential pathogens, all clean metagenomic reads were classified using the MetaPhlAn 4 (v4.1.0) (33) based on the mpa_vJun23_CHOCOPhlAnSGB_202403 database, and only potentially pathogenic microbial species with a relative abundance of >0.001% in at least one sample were retained for further analysis. In addition, after the MAGs assembled in this study were classified based on the GTDB database (reference database version r214), the same strategy was used to identify potential pathogens.

## Metagenomic assembly, genome binning and species-level clustering

All clean reads in each region sample were assembled individually using MEGAHIT (v1.2.9) (61) with default parameters. Genome binning was performed using Meta-BAT2 (v2.12.1) (62) with default parameters. The quality (estimated completeness and contamination) of MAGs was evaluated with CheckM (v1.04) (63) lineage workflow. The non-coding RNA genes were searched using barrnap (v0.9) (64) (https://github.com/tseemann/barrnap) and tRNAscan-SE (v2.0.5) (65). The 30,547 MAGs were clustered at the species-level genome bins (SGBs) using dRep (v2.2.4) (66) with the parameter "-pa 0.9 -sa 0.95 -nc 0.30 -cm larger." Taxonomic annotation of each SGBs was performed with GTDB-Tk (v2.3.2) (42) (reference database version r214) with "classify_wf" workflow using the parameter "--skip_ani_screen."

## Functional annotations and diversity analysis of potentially pathogenic microbial genomes

The protein-coding sequences (CDS) of 1,127 MAGs of potential pathogens were predicted and annotated with Prodigal (v2.6.3) (67) using the parameter "-c -m -p single." ARGs were annotated with the Comprehensive Antibiotic Resistance Database (v3.2.4) (68) using Diamond (v2.0.8.146) (69) blastp with a 95% identity, 90% coverage and e-value of 1e-05 cutoff (26). We manually classified the resistance categories of ARGs and were consistent with previous studies (70). ARGs were manually reorganized according to the antibiotics they resist. Those associated with penam, cephalosporin, carbapenem, cephamycin, penem, and monobactam were consolidated into the beta-lactam category. ARGs linked to macrolides, lincosamides, and streptogramins were grouped into the macrolide-lincosamide-streptogramin category. ARGs conferring resistance to multiple drug classes were placed into the multidrug category. A sequence was annotated as a VFG if it BLASTP search against the Virulence Factor Database (VFDB) (71) met the criteria of e-value 1e−5, 90% similarity, and 90% hit length (72). MRGs were identified by comparison against the BacMet2 database (73) using Diamond (v2.0.8.146) blastp with an e-value 1e−5, a similarity of ≥80%, and coverage of ≥90% (19) and removed antibacterial biocide genes. These parameters have been utilized in previous studies and have proven effective in annotating functional genes from genomes. Moreover, we utilized the mobileOG-db (74) with the parameter "-e 1e-5 --query-cover 90 --id 95" to detect MGEs in MAGs. The ARG, MRG, VFG, and MGE sequences were deduplicated using the "cluster" command of MMseqs2 (v14.7e284) (75) at 95% identity and 90% sequence overlap using the parameter "--min-seq-id 0.95-c 0.9 --cov-mode 0."

The relative abundance of each PARB in each sample was assessed using CoverM (v0.7.0) with parameters "--min-read-percent-identity 95 --min-read-aligned-percent 90 -m tpm." The richness of PARB in 1,382 samples was calculated using the R vegan (v2.6.4) package.

We downloaded 60 reference genomes of the O104:H4 strain from NCBI to compare with the *E. coli* genomes assembled in this study. We randomly selected a genome of *E. coli* O104:H4 and compared the average nucleotide identity with the 90 near-complete

MAGs assembled in this study using the "compare" command of dRep (v2.2.4) (66). The phylogenetic tree of *E. coli* genomes was built using FastTree (v2.1.11) (76) with default settings using the protein sequence alignments generated by GTDB-Tk (v2.3.2) (42). We analyzed the functional genes in near-complete *E. coli* MAGs (completeness >90%) from the gut of eastern and western populations, aiming to minimize errors caused by genome completeness. We used the genome of *Escherichia marmotae* (RefSeq: GCF_029962465.1), which is closely related to *E. coli*, as the outgroup and established the root. Trees were visualized using iTOL(v6) (77).

## Statistical analysis

The R software (v4.2.1) was employed to perform statistical analyses and graphical representations. The R package Nonpareil (v3.4.0) (78) was used to assess the coverage of newly sequenced metagenomic samples. The R package ggmap (v3.0.2) (http://journal.r-project.org/archive/2013-1/kahle-wickham.pdf) was used to visualize the latitude and longitude information of samples on Stamen Maps. Power analysis was performed to estimate the required sample size using R package pwr (v1.3.0). Phylogenetically Independent Contrasts (PIC) analysis was performed to estimate the relationship between ARGs and VFGs, MRGs, and MGEs in genomes of potential pathogens using R package ape (v5.8). Wilcoxon rank-sum test was used to evaluate differences in various variables between the eastern and western populations. Correlations between the richness of potential pathogens and the richness of ARG subtypes in the samples were assessed using Spearman's correlation analysis. Differences in the relative abundance of ARG types or subtypes between groups were analyzed using OmicStudio tools (79) STAMP (80) module by Wilcoxon rank-sum test. Venn diagram and bar plot were performed with EVenn (81) and ImageGP (82). Enrichment analysis of different groups was performed using Fisher's exact test, with $P$ values adjusted ($P_{adj}$) by the False Discovery Rate (FDR) method.

## ACKNOWLEDGMENTS

We would like to thank Yue Jiayu from Wujiaba Primary School, affiliated with Yunnan Normal University, for assisting us in collecting information. We are grateful for support with sample collections by the staff members from the Qiangtang National Nature Reserve, Tibet, China, and the Science and Technology Department of Tibet. We gratefully acknowledge L.X.P. at BGI-Shenzhen for assisting in sample sequencing. The computations were done on the high-performance computers of the Advanced Computing Center of Yunnan University.

This study was supported by the Second Tibetan Plateau Scientific Expedition and Research (STEP) program (no. 2021QZKK0103 and 2019QZKK0503), the Chinese National Natural Science Foundation (no. U2002206 and 31970571), the Yunnan University graduate Research Innovation Project (KC-22221159 and ZC-22221199), the Yunling Scholar of Yunnan Province (Z.Z.), and the central government guides local science and technology development funds (202407AA110009).

## AUTHOR AFFILIATIONS

[1]State Key Laboratory for Conservation and Utilization of Bio-Resources in Yunnan, School of Life Sciences, Yunnan University, Kunming, Yunnan, China
[2]Key Laboratory of Adaptation and Evolution of Plateau Biota, Northwest Institute of Plateau Biology, Chinese Academy of Sciences, Xining, Qinghai, China
[3]College of Animal Science and Technology, Gansu Agricultural University, Lanzhou, Gansu, China
[4]Center for Pan-third Pole Environment, Lanzhou University, Lanzhou, China
[5]Key Laboratory of Pan-third Pole Biogeochemical Cycling, Lanzhou, Gansu Province, China

[6]Chayu Monsoon Corridor Observation and Research Station for Multi-Sphere Changes, Xizang Autonomous Region, Chayu, China

## AUTHOR ORCIDs

Chen Tian ⓘ http://orcid.org/0000-0003-2922-4246
Pengfei Liu ⓘ http://orcid.org/0000-0003-1003-2025
Lei Zhu ⓘ http://orcid.org/0009-0004-3602-1281
Zhigang Zhang ⓘ http://orcid.org/0000-0001-5639-1352

## AUTHOR CONTRIBUTIONS

Chen Tian, Data curation, Formal analysis, Investigation, Methodology, Project administration, Software, Visualization, Writing – original draft | Tongzuo Zhang, Data curation, Writing – review and editing | Daohua Zhuang, Data curation, Methodology, Software | Yu Luo, Data curation, Formal analysis, Methodology | Teng Li, Data curation, Formal analysis, Methodology | Fangfang Zhao, Data curation, Formal analysis | Jianan Sang, Data curation | Zecheng Tang, Investigation, Methodology | Peicheng Jiang, Investigation, Methodology | Tao Zhang, Data curation | Pengfei Liu, Investigation, Writing – review and editing | Lei Zhu, Data curation, Formal analysis, Writing – review and editing | Zhigang Zhang, Conceptualization, Funding acquisition, Supervision, Writing – review and editing

## DATA AVAILABILITY

The clean data and recovered MAGs reported in this paper have been deposited into the CNGB Sequence Archive (CNSA) of China National GeneBank DataBase (CNGBdb) with accession number CNP0005182. The raw sequencing data of the 70 samples are also available at https://db.cngb.org/qtp/. The profiles of gut microbial species and functions and ARGs in eastern and western Chinese populations are deposited in the National Genomics Data Center (NGDC) with project id PRJCA026300. The other data supporting the findings of this study are available within the paper. All mentioned tools used for the data analysis in this study are publicly available, and the version and parameters used have been indicated. The scripts used for metagenomic analysis in this study have been uploaded to https://github.com/Chentian09/Chinese-populations.

## ETHICAL APPROVAL

This study was approved by The Committee on Human Subject Research and Ethics, Yunnan University (Approval No. CHSRE2023004). Informed consent was obtained from all individual participants.

## ADDITIONAL FILES

The following material is available online.

### Supplemental Material

**Supplemental Figures (mSystems01372-24-s0001.pdf).** Fig. S1 to S6.
**Supplemental Tables (mSystems01372-24-s0002.xlsx).** Tables S1 to S15.

### Open Peer Review

**PEER REVIEW HISTORY (review-history.pdf).** An accounting of the reviewer comments and feedback.

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
