## [Reviewer comments · mSystems]

Industrialization drives the gut microbiome and resistome of the Chinese populations

Chen Tian, Tongzuo Zhang, Daohua Zhuang, Yu Luo, Teng Li, Fangfang Zhao, Jianan Sang, Zecheng Tang, Peicheng Jiang, Tao Zhang, Peng-Fei Liu, Lei Zhu, and Zhigang Zhang

Corresponding Author(s): Zhigang Zhang, Yunnan University

Review Timeline:

Submission Date:

October 16, 2024

Accepted:

October 30, 2024

Editor: Sepideh Pakpour

Reviewer(s): Disclosure of reviewer identity is with reference to reviewer comments included in decision letter(s). The following individuals involved in review of your submission have agreed to reveal their identity: Yongqin Liu (Reviewer #3)

Transaction Report:

DOI: <https://doi.org/10.1128/msystems.01372-24>

Re: mSystems01372-24 (Industrialization drives the gut microbiome and resistome of the Chinese populations)

Dear Prof. Zhigang Zhang:

Your manuscript has been accepted, and I am forwarding it to the ASM production staff for publication. Your paper will first be checked to make sure all elements meet the technical requirements. ASM staff will contact you if anything needs to be revised before copyediting and production can begin. Otherwise, you will be notified when your proofs are ready to be viewed.

Sincerely,

Sepideh Pakpour
Editor
mSystems

Reviewer #1 (Comments for the Author):

The authors solved my questions well. The revised version of the manuscript showed an improvement over the previous version. The conclusions of this study imply that lower levels of industrialization do not always mean lower levels of ARGs.

Reviewer #3 (Comments for the Author):

The authors have addressed all my comments.